# Neuroprotective Effect of Dioscin on the Aging Brain

**DOI:** 10.3390/molecules24071247

**Published:** 2019-03-30

**Authors:** Yan Qi, Ruomiao Li, Lina Xu, Lianhong Yin, Youwei Xu, Xu Han, Jinyong Peng

**Affiliations:** 1College of Pharmacy, Dalian Medical University, Western 9 Lvshunnan Road, Dalian 116044, China; friendqy@163.com (Y.Q.); gulinadalian2009@163.com (R.L.); xulina627@163.com (L.X.); yinlianhong1015@163.com (L.Y.); xudlmedu@163.com (Y.X.); Xuhan0118@163.com (X.H.); 2Key Laboratory for Basic and Applied Research on Pharmacodynamic Substances of Traditional Chinese Medicine of Liaoning Province, Dalian Medical University, Dalian 116044, China; 3National-Local Joint Engineering Research Center for Drug Development (R&D) of Neurodegenerative Diseases, Dalian Medical University, Dalian 116044, China

**Keywords:** brain aging, neuroprotection, dioscin, oxidative stress, inflammation

## Abstract

Our previous works have shown that dioscin, a natural product, has various pharmacological activities, however, its role in brain aging has not been reported. In the present study, in vitro H_2_O_2_-treated PC12 cells and in vivo d-galactose-induced aging rat models were used to evaluate the neuroprotective effect of dioscin on brain aging. The results showed that dioscin increased cell viability and protected PC12 cells against oxidative stress through decreasing reactive oxygen species (ROS) and lactate dehydrogenase (LDH) levels. In vivo, dioscin markedly improved the spatial learning ability and memory of aging rats, reduced the protein carbonyl content and aging cell numbers, restored the levels of superoxide dismutase (SOD), glutathione (GSH), glutathione peroxidase (GSH-Px), malondialdehyde (MDA) and nitric oxide synthase (NOS) in brain tissue, and reversed the histopathological structure changes of nerve cells. Mechanism studies showed that dioscin markedly adjusted the MAPK and Nrf2/ARE signalling pathways to decrease oxidative stress. Additionally, dioscin also significantly decreased inflammation by inhibiting the mRNA or protein levels of TNF-α, IL-1β, IL-6, CYP2E1 and HMGB1. Taken together, these results indicate that dioscin showed neuroprotective effect against brain aging via decreasing oxidative stress and inflammation, which should be developed as an efficient candidate in clinical to treat brain aging in the future.

## 1. Introduction

Aging of the population is a common problem around the world. According to data from a 2017 United Nations report, the number of older people aged 60 or over will grow from 962 million to nearly 2.1 billion by 2050 [1]. Brain aging can not only cause the decline of learning and memory ability, but also accelerate the aging process and increase the possibility of suffering from age-related diseases (ARDs) [2,3], which are rapidly becoming a huge strain for medical care and public health systems [4].

Although the pathogenesis of brain aging is very complicated, oxidative stress and inflammation are closely associated with the disease. The brain is particularly susceptible to free radical attack because of its high metabolism and the poor activity of antioxidant enzymes [5]. During aging, neurons tend to accumulate oxidatively-damaged molecules [6,7,8], and excessive reactive oxygen species (ROS) can cause aging and death of nerve cells [9]. In addition, chronic and low-grade inflammation has been observed in the aging brain [10,11]. With age, microglia can exhibit enhanced sensitivity to inflammatory stimuli, and the increased levels of inflammatory cytokines can lead to neurodegeneration [12]. Therefore, regulating oxidative stress and inflammation should be an effective method to treat brain aging.

In recent years, some drugs including rapamycin [13,14,15] and melatonin [16,17] have been used to prevent or delay brain aging and ARDs development by regulating oxidative stress and inflammation, but some studies have suggested that rapamycin has immunosuppressant action, which limits its clinical application [18,19,20]. Thus, it is urgent to find effective drugs with high efficiency and low toxicity to treat brain aging.

Traditional Chinese medicines have been used to treat various diseases in China for a long time. *Dioscoreae rhizoma* (Shanyao in Chinese) is a kind of herbal medicine contained in Shennong’s Classic of Materia Medica, a famous Chinese classical pharmaceutical work, which has the functions of prolonging lifespan. Notably, dioscin (shown in Figure 1A), an active compound from Shanyao, was shown to display potent antioxidant and anti-inflammatory effects [21,22,23,24] with low toxicity [25] in our previous works. However, the protective effect of dioscin on brain aging has not been reported in our best knowledge. The aim of the present study was therefore to investigate the neuroprotective effects and possible mechanisms of action of dioscin in brain aging in vitro and in vivo.

## 2. Results

### 2.1. Dioscin Alleviated H_2_O_2_-Induced Injury in PC12 Cells

As shown in Figure 1B, dioscin at concentrations over 400 ng/mL significantly affected the viability of PC12 cells. Therefore, dioscin at the concentration of less than 400 ng/mL was selected for the subsequent experiments. H_2_O_2_ in the range of 200–500 μM significantly decreased cell viability, which was decreased to 51.87 ± 3.15% by 300 μM of H_2_O_2_ for 2 h treatment (Figure 1C). Therefore, 300 μM of H_2_O_2_ for 2 h was selected to treat PC12 cells.

Under these conditions, dioscin pretreatment significantly restored cell viability in a dose- and time-dependent manner (Figure 1D). Dioscin at the concentrations of 100, 200 and 400 ng/mL for 3 h was selected to protect PC12 cells against H_2_O_2_-induced injury. As shown in Figure 1E, compared with the control group, the lactate dehydrogenase (LDH) level in the model group was significantly increased, and was markedly decreased by dioscin. Furthermore, compared with the control group, the morphological changes, including soma deformation and neurite fracture of PC12 cells caused by H_2_O_2_, were all markedly reversed by dioscin (Figure 1F).

### 2.2. Effects of Dioscin on Body Weight, Blood Glucose and Thymus Coefficient in Rats

As shown in Figure 2A, the body weights of rats in each group increased, but there was no significant difference between them. The body weights of the rats in the 60 mg/kg of dioscin group and vitamin E positive control group were close to those of the control group. Besides, compared with the control group, the blood glucose level in the model group was significantly increased, which was significantly decreased by 60 and 40 mg/kg of dioscin (Figure 2B), indicating that dioscin prevented the hyperglycemia induced by aging. Moreover, compared with the control group, the thymus coefficient in the model group was significantly decreased, indicating that the thymus had obvious atrophy in aging state, which was significantly restored by dioscin (Figure 2C).

### 2.3. Dioscin Improved the Cognition Deficit of Aging Rats

The effects of dioscin on the spatial location ability of aging rats in the Morris water maze (MWM) test are shown in Figure 2D–F.

In the hidden platform experiment, with the extension of training time, the escape latency of rats in each group showed a downward trend. Compared with the control group, the escape latency was significantly prolonged in the model group (Figure 2D), suggesting obvious spatial learning and memory disorders in aging rats. Compared with the model group, the levels of escape latency in the dioscin-treated groups were significantly shortened on the second day, and the levels were basically stable over the next days. In the probe trial (Figure 2E), compared with the control group, the platform crossing times and the time spent in the target platform quadrant in the model group were significantly reduced (*p* < 0.05), which was reversed by dioscin with *p* < 0.01. In terms of search strategy, the search strategy of the model group was mainly random and aimless, while the data in the dioscin groups showed a straight line and purpose, suggesting the embodiment of strong spatial learning and memory ability (Figure 2F). In a word, the above results indicated that dioscin significantly improved the cognition deficit of aging rats.

### 2.4. Dioscin Attenuated Neuropathological Changes in Aging Brain

As shown in Figure 3, the rats in model group showed that the cytoplasm of neurons in cortex was understain and the array of hippocampal neurons was disordered and incompact. The cortical and hippocampal morphologies of dioscin groups were restored, suggesting the effect of dioscin on reducing the neuron damages of aging brain. Immunofluorescence assay of brain sections in model group showed that there were less microtubulin-associated protein 2 (MAP2)-positive neurons, more glial fibrillary acidic protein (GFAP)-positive astrocytes and ionized calcium binding adapter molecule 1 (Iba1)-positive microglias (Figure 4) in aging brain than those of in control group, which were obviously reversed by dioscin. The above results indicated that dioscin played neuroprotective effects by improving the structure of neurons synapses and inhibiting the activation of astrocytes and microglias.

### 2.5. Dioscin Attenuated Ultrastructure Changes in Aging Brain

Transmission electron microscopy (TEM) assays (Figure 5) showed that the neurons in the control group presented intact nuclear membranes (green arrows), normal nuclei, and numerous mitochondria (red arrows) with clear cristae. Compared with the control group, the neurons in the model group exhibited blurred nuclear membranes, mitochondria with swelling degeneration, fractured cristae and relatively reduced endoplasmic reticulum. Dioscin (60 mg/kg) significantly improved the ultrastructure of the aging brain by conserving intact nuclear membranes with double layer structures, and mitochondria with normal shapes and structures. The results indicated that dioscin improved the ultrastructure of neurons in aging rats to exert its neuroprotective effect.

### 2.6. Dioscin Alleviated Oxidative Damage In Vitro and In Vivo

β-Galactosidase staining experiments results (Figure 6A) show that the numbers of blue senescent cells in the model group were obviously increased, which was significantly reversed by dioscin. Protein carbonylation has been widely used to measure oxidative damage. As shown in Figure 6B, compared with the control group, the protein carbonyl contents in the model groups were significantly increased, which was markedly reduced by dioscin both in H_2_O_2_-injured PC12 cells and aging rats. The above results suggest that dioscin displays neuroprotective effects through reducing protein carbonyl content and cellular senescence. As shown in Figure 6C, compared with the control group, the ROS level in the model group was significantly increased, which was markedly reversed by dioscin. Moreover, as shown in Figure 6D, compared with model group, dioscin significantly improved the levels of SOD, GSH and GSH-Px, and reduced MDA and NOS levels in brain tissue. These results indicate that dioscin significantly alleviates brain aging via suppressing oxidative stress.

### 2.7. Dioscin Adjusted Nrf2/ARE Signal Pathway

The data in Figure 7 display that the expression levels of Nrf2, NQO1, HO1, GST and SOD1 in model group were significantly down-regulated, and the expression level of Keap1 was significantly up-regulated compared with control group, and these effects were significantly reversed by dioscin in vitro and in vivo. These results suggested that the effects of dioscin against brain aging should be through modulating Nrf2/ARE signalling.

### 2.8. Dioscin Attenuated MAPKs Phosphorylation

The results in Figure 8 show that dioscin markedly inhibited the MAPK signaling pathway by down-regulating the levels of p-ERK/ERK, p-p38/p38 and p-JNK/ JNK in vitro and in vivo.

### 2.9. Dioscin Ameliorated Neuroinflammation

As shown in Figure 9A,B, dioscin significantly down-regulated the expression levels of CYP2E1 and HMGB1 compared with the model groups. Additionally, increased mRNA levels of tumor necrosis factor-α (TNF-α), interleukin-1β (IL-1β) and interleukin-6 (IL-6) were observed in the model groups, which were also markedly decreased by dioscin (Figure 9C). These results indicated that dioscin delayed brain aging through ameliorating neuroinflammation.

## 3. Discussion

Brain aging is often accompanied by a series of structural and functional changes which will not only lead to a decline of cognitive functions, but also affect the quality of life [26,27,28]. In this study, our results indicated that dioscin markedly improved the structure and function of the aging brain by decreasing oxidative stress and inflammation, and restoring histopathological brain changes.

Aging and ARDs can be caused in various cell components by free radicals [6,29], and the physiological and histological characteristics of the brain determine that it is particularly vulnerable to free radical-oxidative damage [5]. The purpose of this study was to investigate the neuroprotective effect of dioscin on brain aging. In vitro, a model of PC12 cells treated by H_2_O_2_ and a d-galactose-induced brain senescence model in rats were used. It is known that an exogenous dose of d-galactose beyond normal concentrations can induce brain aging by increasing oxidative stress, apoptosis and inflammation. d-galactose-injected rodent models recapitulate many features of brain ageing which have been extensively used for pharmacology and mechanistic studies of brain aging [30,31].

In brain aging, some cognitive functions including memory, reasoning and spatial ability may decline, and oxidative damage plays a major role in the disorder [32,33]. Behavior tests can reflect the changes in learning and memory functions of animals. At present, the Morris water maze is recognized as an objective evaluation method for learning and memory function [34]. In this study, a Morris water maze test was used to evaluate the effects of dioscin on spatial memory and learning ability of aging rats. The results showed that dioscin significantly improved the cognitive dysfunctions caused by brain aging.

The cognitive changes of brain aging are related to changes in the cerebral cortex, hippocampus and synaptic interface. Neurons and glial cells are the main cells in the central nervous system. In this study, nissl staining, immunofluorescence and transmission electron microscopy assays were used to investigate the effects of dioscin on the morphology and structure of neurons, astrocytes and microglia in the aging brain. Neurons are the main basic components to realize the complex functions of the brain. With the increase of age, the numbers of synapses will not only decrease, but also have a tendency to shorten with age. The change or loss of structure and function of synapses is an important factor leading to brain aging [35,36]. Our results showed that dioscin improved the neuron structure damage caused by aging via maintaining the normal morphology of neurons, restoring the numbers and length of synapses, and changing the ultrastructure of aging neurons against oxidative damage.

Glial cells, including microglia and astrocytes, are the major mediators of neuroinflammation. Activated microglial cells can trigger inflammatory responses, and neuroinflammation may contribute to the pathogenesis of many central nervous system disorders, including brain aging and ARDs [37,38]. As the specific markers of astrocytes and microglia, the expression levels of GFAP and Iba1 were increased in aging groups, indicating that astrocytes and microglia were activated and neuroinflammation occurred. In this study, we found that the increased expression levels of GFAP and Iba1 were significantly reversed by dioscin, indicating that dioscin showed protective effects against neuroinflammation.

Free radical damage and oxidative stress play important roles in nervous system diseases. Mitochondria are the main site of ROS production. Once ROS metabolism is out of balance, mitochondrial structure and function will be damaged, and neurons will eventually die. In this study, a DCFH-DA probe was used to detect ROS levels, as intracellular ROS can oxidize non-fluorescent DCFH to generate fluorescent DCF. The fluorescence intensity is proportional to the ROS level. As mentioned in the TEM results, the shape and structure of mitochondria in the aging brain were damaged, which was obviously reversed by dioscin. Our work indicated that dioscin reduced ROS generation, decreased the production of oxidative damage products, and improved the levels of antioxidant enzymes, including SOD and GSH-Px, to restore the metabolic balance of mitochondrial ROS. This result was consistent with the changes in mitochondrial structure indicated in the TEM assay suggesting that the improvement of mitochondrial structure by dioscin was via inhibiting ROS accumulation. Basis on these results, we conclude that the neuroprotevtive effect of dioscin on the aging brain may be through suppressing oxidative stress and improving mitochondrial dysfunction.

Nuclear factor erythroid 2 related factor 2 (Nrf2) has been classically considered as a major regulator of the antioxidant response, and Kelch-like Ech-associated protein-1 (Keap1) is a negative regulator of Nrf2 [39,40]. When cells are stimulated by oxidative stress, Nrf2 will uncouple from Keap1 and transfer to the nucleus, where it can bind with antioxidant response elements (ARE) by up-regulating numerous cytoprotective genes, including heme oxygenase-1 (HO1), glutathione-S- transferase (GST), NAD(P)H quinone oxidoreductase 1 (NQO1) [40,41]. Our data showed that dioscin significantly down-regulated the expression level of Keap1, and up-regulated the expression levels of Nrf2, NQO1, HO1, GST and SOD1 compared with the model groups in vitro and in vivo. These results suggest that the neuroprotective effects of dioscin against brain aging should be through adjusting Nrf2/ARE signalling.

A previous study has demonstrated that mitogen-activated protein kinase (MAPK) signaling is involved in the regulation of Nrf2-mediated oxidative stress in aging rats [42]. MAPKs include the extracellular signal-regulated kinases 1 and 2 (ERK1/2), c-Jun amino-terminal kinases 1 to 3 (JNK1 to 3) and p38 families which can regulate diverse cellular programs by relaying extracellular signals to intracellular responses. Some factors, including cytokines and growth factors, can activate MAPKs and participate in the regulation of oxidative stress, inflammation and apoptosis [43,44]. The results present here showed that dioscin markedly attenuated MAPK signal by down-regulating the levels of p-ERK/ERK, p-p38/p38 and p-JNK/JNK in vitro and in vivo. We confirmed that the anti- aging effect of dioscin on brain was via adjusting the MAPK pathway.

In the aging brain, chronic and low-grade inflammation has been observed [10,11]. In addition, oxidative stress can induce MAPK phosphorylation, and increase the expression levels of the proinflammatory cytokines, including IL-1β, IL-6 and TNF-α [45]. High mobility group box protein 1 (HMGB1), a ubiquitous nuclear protein released by glia and neurons upon inflammasome activation, is a key initiator of neuroinflammation [46]. Cytochrome P450-2E1 (CYP2E1) has been involved in the neurophysiology which can exacerbate neurological deficit and increase ROS formation, oxidative stress, inflammation, and neurodegeneration [47,48]. Our results demonstrated that dioscin markedly decreased the mRNA or protein levels of TNF-α, IL-1β, IL-6, CYP2E1 and HMGB-1 in vitro and in vivo. These results suggested that the protective effects of dioscin against brain aging are through ameliorating neuroinflammation.

## 4. Conclusions

Taken together, this study demonstrated that dioscin showed potent effects against brain aging via decreasing oxidative stress and inflammation. These findings suggest that dioscin should be developed as an efficient candidate in clinical to treat brain aging in the future. We also suggest that patients with brain aging should take Shanyao for therapy and health. Of course, further studies are still essential to elucidate the underlying mechanisms.

## 5. Materials and Methods

### 5.1. Chemicals and Reagents

Dioscin (purity > 98%) was prepared in our laboratory [49]. It was dissolved in 0.5% carboxymethylcellulose sodium (CMC-Na) for in vivo experiments and in 0.01% dimethylsulfoxide (DMSO) for in vitro tests. Vitamin E used as the positive control drug was procured from Hangzhou Huadong Medicine Group Wufeng Pharmaceutical (Hangzhou, China), and was dissolved in oil for in vivo experiments. d(+)Galactose was purchased from Solarbio Technology (Beijing, China). 3-(4, 5-Dimethylthiazol-2-yl)-2,5-diphenyltetrazolium bromide (MTT) was provided by Roche Diagnostics GmbH (Mannheim, Germany). H_2_O_2_ was provided by Kermel Chemical Reagent (Tianjin, China). Lactate dehydrogenase (LDH), malondialdehyde (MDA), nitric oxide synthase (NOS), superoxide dismutase (SOD), glutathione (GSH) and glutathione peroxidase (GSH-Px) assay kits were obtained from the Nanjing Jiancheng Institute of Biotechnology (Nanjing, China). The protein carbonyl assay kit was obtained from Comin Biotechnology (Suzhou, China). The senescence β-galactosidase staining kit, tissue protein extraction kit, bicinchoninic acid (BCA) protein assay kit and ROS assay kit were obtained from Beyotime Biotechnology (Jiangsu, China). 4,6-Diamidino-2- phenylindole (DAPI) was purchased from Sigma Chemical (St. Louis, MO, USA). Antibodies were obtained from Proteintech Group (Chicago, IL, USA) and Bioworld Technology (St. Louis Park, MN, USA). RNAiso Plus, PrimeScript^®^ RT reagent Kit with gDNA Eraser, SYBR^®^
*Premix Ex Taq*™ II (Tli RNaseH Plus) and the primers were purchased from TaKaRa Biotechnology (Dalian, China).

### 5.2. Cell Culture

PC12 cell line was obtained from Shanghai Institutes for Biological Sciences, Chinese Academy of Sciences (Shanghai, China). The cells were cultured in RPMI 1640 medium (Hyclone, South Logan, UT, USA), containing 10% fetal bovine serum (FBS, Hyclone) at 37 °C with 5% CO_2_.

### 5.3. Cytotoxicity of Dioscin

The cells were seeded into 96-well plates at a density of 1 × 10^4^ cells per well and incubated for 24 h, and then treated with different concentrations dioscin (1600, 800, 400, 200, 100, 50 and 25 ng/mL) for 24 h. Cell viability was measured by the MTT assay. A total of 10 μL of MTT solution (5 mg/mL) was added to each well for 4 h incubation at 37 °C, and then 150 μL of DMSO was added to dissolve formazan crystals. Absorbance of each well was measured at 490 nm using a microplate reader (Thermo, Waltham, MA, USA).

### 5.4. H_2_O_2_-Induced Cell Injury

The cells were seeded into 96-well plates at a density of 1 × 10^4^ cells per well and incubated for 24 h, and then treated with different concentrations of H_2_O_2_ (200, 300, 400, and 500 μM) for 1, 2, 3 and 4 h. Cell viability was measured by MTT assay as described above. The suitable concentration and treatment time of H_2_O_2_ in vitro experiments were optimized according to the result.

### 5.5. Cell Viability Study

The cells were seeded into 96-well plates at a density of 1 × 10^4^ cells per well and incubated for 24 h. The cells in treatment groups were pretreated with different concentrations of dioscin (400, 200, 100, 50, 25 and 12.5 ng/mL, final concentration) for 1, 2 and 3 h. Except for the control group, the cells in other groups were treated with 300 μM of H_2_O_2_ for 2 h. Effects of dioscin on cell viability were evaluated by the MTT assay. Moreover, the cell morphologies were imaged using a phase contrast microscope (TS100, Nikon, Tokyo, Japan).

### 5.6. Detection of LDH Release and Intracellular Protein Carbonyl

PC12 cells were seeded into 96-well plates at a density of 1 × 10^4^ cells per well and incubated for 24 h. Then, the medium of control group and model group were replaced with serum-free RPMI 1640, while the cells in treatment groups were treated with dioscin (400, 200 and 100 ng/mL, final concentration) for 3 h. Except for the control group, the cells in other groups were treated with 300 μM of H_2_O_2_ for 2 h. The release of LDH and the content of intracellular protein carbonyl were assayed by the kits.

### 5.7. Cellular Senescence β-galactosidase Staining

The cells were plated in 6-well plates at a density of 2 × 10^5^ cells/well and incubated for 24 h. Then, the medium of control group and model group were replaced with serum-free RPMI 1640, while the cells in treatment groups were treated with dioscin (400, 200 and 100 ng/mL, final concentration) for 3 h. Except for the control group, the cells in other groups were treated with H_2_O_2_ (300 μM) for 2 h. The senescent cells were stained by senescence β-galactosidase staining kit. Moreover, the images were obtained by using a phase contrast microscope (TS100, Nikon).

### 5.8. Determination of Intracellular ROS Level

The culture and treatment of PC12 cells were as the same as cellular senescence β-galactosidase staining. After removing cell culture medium, 1 mL of DCFH-DA (10.0 μM) was added into the well and incubated for 20 min at 37 °C. The samples were photographed with an inverted fluorescence microscope (TE2000U, Nikon).

### 5.9. Animals and Treatment

Sixty male Wistar rats weighting 280 ± 30 g (3-month old) were obtained from the Experimental Animal Center at Dalian Medical University (Dalian, China) (SCXK: 2013-0006). The animal experiments were approved by the Animal Care & Welfare Committee of Dalian Medical University and all experimental procedures were performed according to the ethical principles of experimental animal welfare. All animals were maintained in a controlled environment with a 12/12 h light schedule, constant temperature (20 ± 3 °C), and relative humidity (60 ± 10%). The rats had free access to food and water. The rats were randomly divided into six groups (n = 10), including control group, model group, dioscin-treated groups at the doses of 60, 40 and 20 mg/kg, and positive control group (27 mg/kg vitamin E). Normal saline was injected intraperitoneally once daily to the rats in control group, while d-galactose (120 mg/kg) was given to other animals. One hour later, vehicle (0.5% CMC-Na) was intragastrically administrated to the rats in control and model groups, and the tested drugs were given to the treated animals once daily for 8 weeks. Within 24 h after the last day, behavioral changes in each group were detected by Morris water maze test. Twenty-four hours later, blood glucose level was measured. Finally, the animals were sacrificed and their brains and thymus glands were collected. The thymus glands were weighed to calculate the organ coefficient and the brain tissues were stored for further assays.

### 5.10. Behavioral Test by Morris Water Maze

Spatial learning and memory function of rats was assessed using a Morris water maze (MWM) test [34]. The MWM system (Institute of Materia Medica, Chinese Academy of Medical Sciences, Beijing, China) consisted of a black circular pool (150 cm diameter × 50 cm height) filled with water (23 ± 2 °C and 24 cm depth) and a video tracking analysis system. A circular platform (10 cm diameter × 23 cm height) was located inside the pool. The rats were received hidden platform test for 4 consecutive days and probe trial on Day 5. For the hidden platform test, the rats were trained once in each quadrant into the water per day. In each trial, they were given 60 s to find the hidden platform, and each one was allowed to remain on the platform for 5 s. When the rats climbed onto the platform, the test terminated and the escape latency was recorded. If a rat failed to find the platform in 60 s, it was guided onto the platform and remained on it for 10 s. In the probe trial on day 5, the platform was removed, and each rat was allowed to explore the pool freely for 60 s. The parameters including swimming trajectory, times of crossing the plat and the time spent in the target platform quadrant were recorded and analyzed by the video tracking analysis system.

### 5.11. Determination of Protein Carbonyl, MDA, NOS, SOD, GSH and GSH-Px Levels in Brain Tissue

The content of protein carbonyl in brain tissue was detected using the assay kit according to the instructions. The cerebral cortex tissues were taken to prepare 10% tissue homogenates. Then, the supernatant were obtained through centrifuging at 2500× *g* for detecting the levels of MDA, NOS, SOD, GSH and GSH-Px using the assay kits.

### 5.12. Nissl Staining

Nissl body is one of the characteristic structures of neurons. The brain tissues were fixed in 10% formalin, embedded in paraffin and sectioned into 5 μm slices. Then, the slices were stained with Nissl solution and the images were obtained using an inverted microscope (TE2000U, Nikon) with 400× magnification.

### 5.13. Senescence Related β-galactosidase Staining In Vivo

Senescence related β-galactosidase is a widely used biological marker of aging. The paraffin sections of brain were dewaxed and hydrated in a conventional manner. Then, the sections were stained by senescence β-galactosidase staining kit. Moreover, the images were obtained by using an inverted microscope (TE2000U, Nikon) with 400× magnification.

### 5.14. Transmission Electron Microscopy Assay

Fresh cortex and hippocampus tissues (control, model and dioscin 60 mg/kg group) were fixed in 2% glutaraldehyde overnight at 4 °C. Then, the samples were cut into tissue blocks about 1 mm^3^ and treated as previously described [50]. The ultramicrotomies were stained and photographed with a transmission electron microscope (JEM-2000EX, JEOL, Tokyo, Japan) with 40,000× magnification.

### 5.15. Immunofluorescence Assay

Microtubulin-associated protein 2 (MAP2), glial fibrillary acidic protein (GFAP) and ionized calcium binding adapter molecule 1 (Iba1) are specific markers of neurons, astrocytes and microglia, respectively. The numbers of positive cells were used to evaluate the neuroprotective effects of dioscin against aging in rats. The paraffin sections of brain were dewaxed and hydrated in a conventional manner, and then treated with citrate buffer (pH = 6.0) in a microwave oven for thermal repair for 20 min. Next, the sections were incubated in a humidified box at 4 °C overnight with anti-MAP2, anti-GFAP or anti-Iba1 (1:70, dilution) antibodies. Finally, the sections were incubated with an alexa fluorescein-labeled secondary antibody for 1 h at 37 °C and counterstained with DAPI (1.0 μg/mL) for 10 min. The images were obtained by fluorescence microscopy (TE2000U, Nikon) with 200× magnification.

### 5.16. Quantitative Real-Time PCR Assay

Total RNA samples from cells and cortex tissues of rats were obtained using RNAiso Plus Kit. RNA samples were reverse transcribed into cDNA using PrimeScript^®^ RT reagent Kit with gDNA Eraser (Perfect Real Time). The levels of mRNA expression were quantified by SYBR^®^
*Premix Ex Taq*™ II (Tli RNaseH Plus) by ABI 7500 Real Time PCR System (Applied Biosystems, Carlsbad, CA, USA). The forward and reverse primers used in the present study are presented in Table 1.

### 5.17. Western Blot Assay

Total protein samples from cells and cortex tissues of rats were extracted and the protein content was determined. Proteins were subjected to SDS-PAGE (10–15%) and then were transferred to PVDF membranes (Millipore, Burlington, MA, USA). After blocking nonspecific binding sites with 5% dried skim milk, the membranes were incubated overnight at 4 °C with primary antibodies (Table 2). The blots were incubated with horseradish peroxidase-conjugated secondary antibody at room temperature for 3 h. Protein expression was detected using enhanced chemiluminescence method by Bio-spectrum Gel Imaging System (UVP, Upland, CA, USA). The relative protein expression was normalized with GAPDH as an internal control.

### 5.18. Statistical Analysis

All numerical data were expressed as the mean ± standard deviation (SD) and analyzed using SPSS 17.0 software (IBM, Armonk, NY, USA). Significant differences among multiple groups were analyzed by one-way ANOVA test followed by LSD test. The results were considered to be statistically significant at *p* < 0.05.

## Figures and Tables

**Figure 1 molecules-24-01247-f001:**
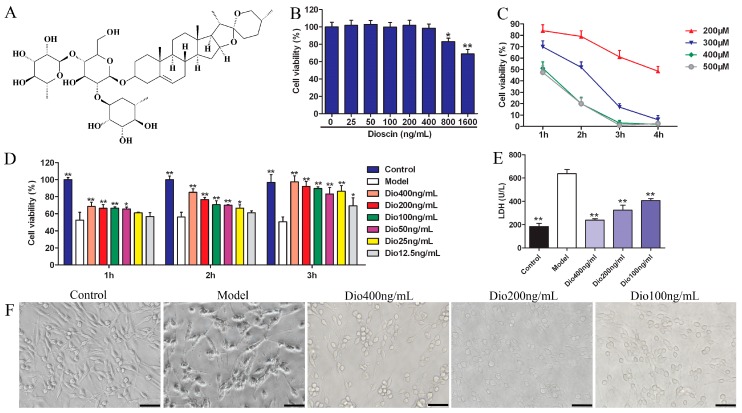
Protective effects of dioscin on H_2_O_2_-induced injury in PC12 cells. (**A**) The chemical structure of dioscin. (**B**) Cytotoxicity of dioscin on PC12 cells detected by MTT assay. (**C**) Effects of H_2_O_2_ on cell viability. (**D**) Effects of dioscin alleviated on H_2_O_2_-induced cell injury. (**E**) Effects of dioscin on LDH release in H_2_O_2_-injured PC12 cells. (**F**) Effects of dioscin on the morphological changes of PC12 cells after H_2_O_2_ treatment (400× magnification; Scale bar = 50 μm). Data are presented as the mean ±SD (n ≥ 5). * *p* < 0.05 and ** *p* < 0.01 compared with model group.

**Figure 2 molecules-24-01247-f002:**
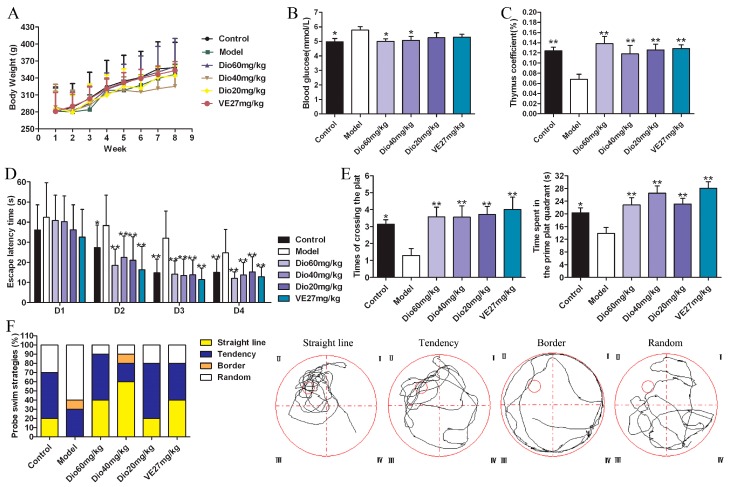
Effects of dioscin on d-galactose induced aging in rats. (**A**) Effects of dioscin on body weights. (**B**) Effects of dioscin on blood glucose. (**C**) Effects of dioscin on thymus gland. (**D**) Effects of dioscin on aging rats in the hidden platform experiment in the WMW test. (**E**) Effects of dioscin on aging rats in the probe trial in the WMW test. (**F**) Effects of dioscin on terms of search strategy in the WMW test. Data are presented as the mean ±SD (n = 10). * *p* < 0.05 and ** *p* < 0.01 compared with model group.

**Figure 3 molecules-24-01247-f003:**
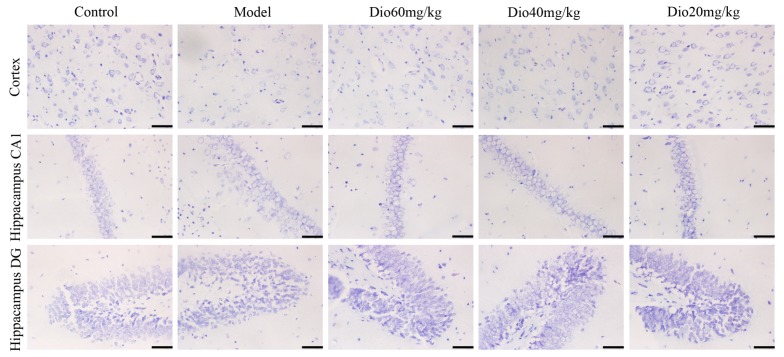
Effects of dioscin on Nissl staining of cortex and hippocampus in aging rat brains (400× magnification). Scale bar = 50 μm.

**Figure 4 molecules-24-01247-f004:**
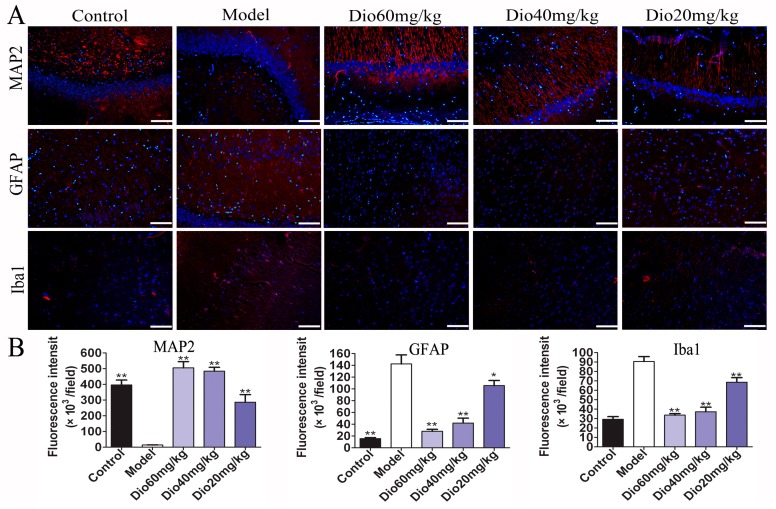
Effects of dioscin on neuropathological changes in aging brain. (**A**) Effects of dioscin on the expression levels of MAP2, GFAP and Iba1 based on immunofluorescence staining (200× magnifica- tion; Scale bar = 100 μm). (**B**) Statistical analyses of fluorescence intensity of the positive cells in rat brains. Data are presented as the mean ±SD (n = 10). * *p* < 0.05 and ** *p* < 0.01 compared with model groups.

**Figure 5 molecules-24-01247-f005:**
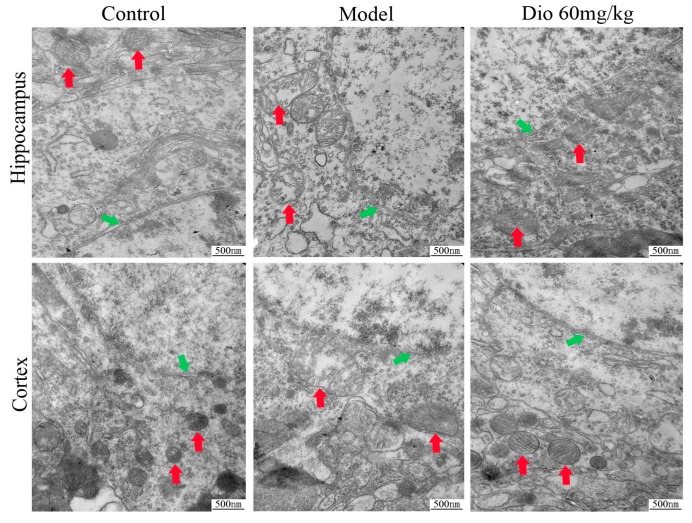
Effects of dioscin on ultrastructure changes in aging brain by transmission electron microscopy (TEM) assay (40,000× magnification). Nuclear membrane and mitochondria are indicated by green arrows and red arrows, respectively.

**Figure 6 molecules-24-01247-f006:**
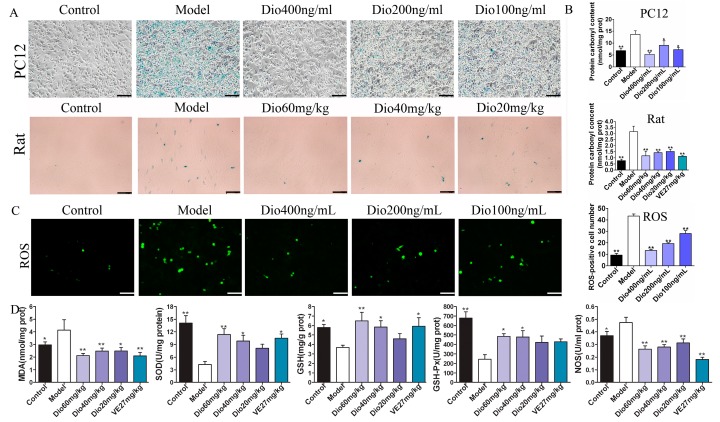
Effects of dioscin on oxidative damage and cell senescence in vitro and in vivo. (**A**) Dioscin reduces cell senescence in vitro (200× magnification, Scale bar = 100 μm) and in vivo (400× magnification; Scale bar = 50 μm). (**B**) Dioscin reduces protein carbonyl content in vitro and in vivo. (**C**) Effects of dioscin on intracellular ROS level in PC12 cells treated by H_2_O_2_ (200× magnification; Scale bar = 100 μm). (**D**) Effects of dioscin on the levels of malondialdehyde (MDA), superoxide dismutase (SOD), glutathione (GSH), glutathione peroxidase (GSH-Px) and nitric oxide synthase (NOS) in brain tissue of aging rats. Data are presented as the mean ±SD (n = 10). * *p* < 0.05 and ** *p* < 0.01 compared with model groups.

**Figure 7 molecules-24-01247-f007:**
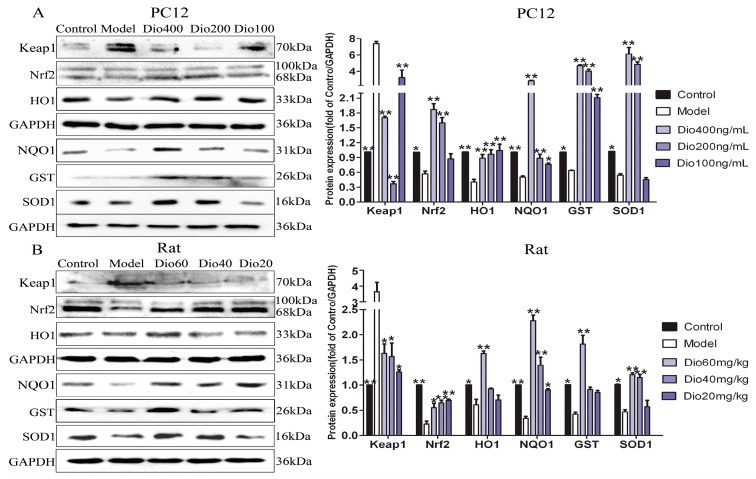
Dioscin-adjusted Nrf2/ARE signal pathway. (**A**) Effects of dioscin on the expression levels of keap1, Nrf2, HO-1, NQO1, GST and SOD1 in PC12 cells treated by H_2_O_2_. (**B**) Effects of dioscin on the expression levels of keap1, Nrf2, HO-1, NQO1, GST and SOD1 in brain tissue of aging rats. Data are presented as the mean ±SD (n = 3). * *p* < 0.05 and ** *p* < 0.01 compared with model groups.

**Figure 8 molecules-24-01247-f008:**
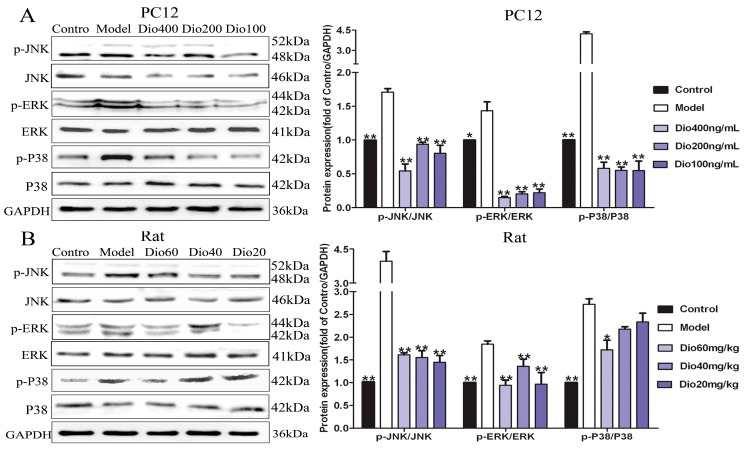
Dioscin adjusted MAPKs signal pathway. (**A**) Effects of dioscin on the levels of phosphorylated MAPKs in PC12 cells treated by H_2_O_2_. (**B**) Effects of dioscin on the levels of phosphorylated MAPKs in brain tissue of aging rats. Data are presented as the mean ±SD (n = 3). * *p* < 0.05 and ** *p* < 0.01 compared with model groups.

**Figure 9 molecules-24-01247-f009:**
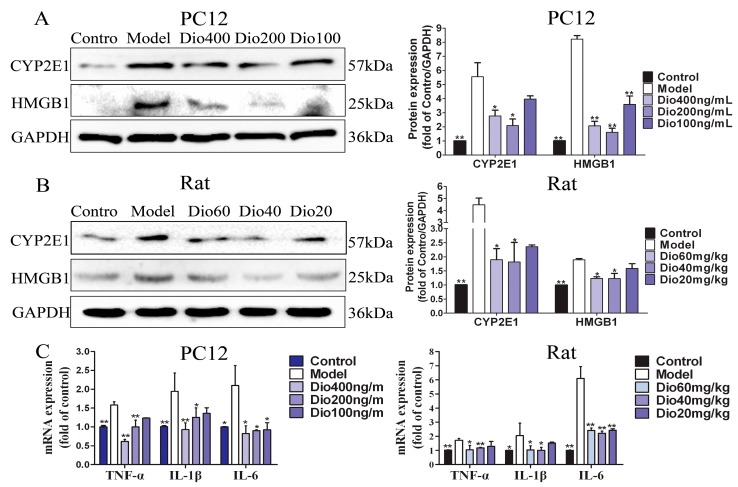
Effects of dioscin on neuroinflammation. (**A**) Effects of dioscin on the expression levels of CYP2E1 and HMGB1 in PC12 cells treated by H_2_O_2_. (**B**) Effects of dioscin on the expression levels of CYP2E1 and HMGB1 in brain tissue of aging rats. (**C**) Effects of dioscin on the mRNA levels of TNF-α, IL-1β and IL-6 in vitro and *in vivo.* Data are presented as the mean ±SD (n = 3). * *p* < 0.05 and ** *p* < 0.01 compared with model groups.

**Table 1 molecules-24-01247-t001:** The sequences of the primers used for real-time PCR assay.

Gene	Forward Primer (5′-3′)	Reverse Primer (5′-3′)
TNF-α	TCAGTTCCATGGCCCAGAC	GTTGTCTTTGAGATCCATGCCATT
IL-1β	CCCTGAACTCAACTGTGAAATAGCA	CCCAAGTCAAGGGCTTGGAA
IL-6	ATTGTATGAACAGCGATGATGCAC	CCAGGTAGAAACGGAACTCCAGA
GAPDH	GGCACAGTCAAGGCTGAGAATG	ATGGTGGTGAAGACGCCAGTA

**Table 2 molecules-24-01247-t002:** The information of the antibodies used in the present work.

Antibody	Separation Gel Concentration (%)	Dilution	Company
GAPDH	10	1:5000	Proteintech Group, Chicago, IL, USA
p38	10	1:500	Bioworld Technology, St. Louis Park, MN, USA
p-p38	10	1:500	Bioworld Technology, St. Louis Park, MN, USA
ERK	10	1:500	Bioworld Technology, St. Louis Park, MN, USA
p-ERK	10	1:500	Bioworld Technology, St. Louis Park, MN, USA
JNK	10	1:500	Bioworld Technology, St. Louis Park, MN, USA
p-JNK	10	1:500	Bioworld Technology, St. Louis Park, MN, USA
CYP2E1	10	1:1000	Proteintech Group, Chicago, IL, USA
HMGB1	12	1:1000	Proteintech Group, Chicago, IL, USA
Keap1	10	1:1000	Proteintech Group, Chicago, IL, USA
Nrf2	10	1:1000	Proteintech Group, Chicago, IL, USA
HO1	10	1:500	Proteintech Group, Chicago, IL, USA
GST	10	1:500	Proteintech Group, Chicago, IL, USA
NQO1	10	1:1000	Proteintech Group, Chicago, IL, USA
SOD1	15	1:1000	Proteintech Group, Chicago, IL, USA

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
