# Peer review of "Neuroprotective Effect of Dioscin on the Aging Brain"

_molecules, 2019, doi:10.3390/molecules24071247_

Round 1

Reviewer 1 Report

Please edit the manuscript for english grammar. 

Figure 1 please indicate the method by which cell viability was measured in the figure legend and results text. 

For figures with microscope images scale bars need to be included. For figure 4 images should show the same area of the brain. It appears some images are showing the hippocampal dentate gyrus while others it is unclear what the structure shown is based on DAPI staining. 

Providing quantitative data for figure 5 would be helpful. 

Figure 7A PC12 cell western blot labels for the concentration of Dio compound are not equivalent to what is reported in the graph or the text of the manuscript. 

Some figures are blurry and difficult to read, in particular figure 9 text on the graphs is too small.

Author Response

Replaying to the comments from Reviewer 1:

1, Please edit the manuscript for english grammar.

Response: Thank you for the valuable comment. The grammar mistakes have been carefully corrected and the whole manuscript has been checked by an English native speaker.

Figure 1 please indicate the method by which cell viability was measured in the figure legend and results text.

Response: Thank you for giving us the valuable comment. Cell viability was measured by MTT assay. Based on your comment, the method has been indicated in the figure legend and results text on page 7 (colored in red).

For figures with microscope images scale bars need to be included. For figure 4 images should show the same area of the brain. It appears some images are showing the hippocampal dentate gyrus while others it is unclear what the structure shown is based on DAPI staining.

Response: Thank you for giving us the valuable comment. Scale bars have been inserted in all microscope figures. In Figure 4, images of hippocampal CA area were selected to show the changes of MAP2 of neurons. Images of cortex area were selected to show the changes of GFAP of astrocytes and Iba1 of microglia. In addition, the images have been quantified. Please see Figure 4 on page 9.

Providing quantitative data for figure 5 would be helpful.

Response: Figure 5 showed the results of TEM assay and it seems hard to quantify. If your comment is for Figure 4, it has been quantified base on your comment.

Figure 7A PC12 cell western blot labels for the concentration of Dio compound are not equivalent to what is reported in the graph or the text of the manuscript.

Response: Thank you for giving us the valuable comment. The right concentration of dioscin compounds have been labeled in Figure 7A PC12 cell western blot labels based on your comment. Please see Figure 7A on page 11.

Some figures are blurry and difficult to read, in particular figure 9 text on the graphs is too small.

Response: Thank you for the valuable comment. Figures have been edited again to improve picture quality and the text on graphs has been enlarged for reading.

Reviewer 2 Report

The current manuscript entitled “Protective Effect of Dioscin on Brain Aging in Rats” by Qi and coworkers deals with the eventual neuroprotective effects of dioscin, a natural steroidal saponin found in several vegetal species, against brain aging. To this aim, the authors have used different experimental models both in PC12 cell culture and in rats.  This paper is continuation of numerous previous works of the same research group, in which the authors have repeatedly shown a protective effect of dioscin in a wide range of pathologies.

Although for the aforementioned reason, the novelty of this study is questionable, the topic of this study is an important one, since aging of population is one of the main challenges of the humanity for the next decades. Therefore, the quest for nootropic agents or brain aging delaying drugs is an intense subject in neuropharmacology nowadays

According to the results found by the authors, dioscin, both in vitro and in vivo, had powerful neuroprotective effects, preventing or reducing several neurochemical markers of brain damage, including ROS, expression of proteins involved in neuronal integrity, histological and ultrahistological markers and even having a noticeable impact on learning and memory.

The number of experiments carried out by the authors in order to determine the beneficial effects of dioscin is outstanding and impressive. Finally, the authors conclude that dioscin-mediated effects are associated to a reduction of oxidative stress by normalizing the Nrf2/ARE signal pathway.

I do really consider that the present study is well performed and soundness. Nevertheless, there are some concerns that the authors should address. My main concern is how the different models resemble the real senescence process.

Major comments:

1.       Although the manuscript is understandable, the writing has too many mistakes and weird expressions. Therefore, I highly recommend to check the whole manuscript by an English native speaker.

2.       Authors have to discuss the neuroprotective effect of dioscin under the light that several previous studies (some from your research group) have showed that dioscine promotes apoptosis and cell death. Those previous results pointed to dioscin as an eventual cytotoxic agent against multiple types of cancer.

3.       All studies carried out point to a general neuroprotective effect of dioscin, but not an anti-aging effect. Although, the D-galactose rat model is a well-recognized model of aging, resembling many of the features of aged rats, I really feel that the authors should have used normal aged rats to test the putative anti-aging effects of dioscin.  Indeed, in order to claim antiaging actions of dioscin, a longitudinal study in normal animals   Moreover, the in vitro study (H2O2-induced damage in PC12 cell line) is not consider a model of aging at all. Rather, it is a model of cell damage mediated by oxidative stress, which is obviously is found in elderly but also in many other brain pathologies.

For this reason, I think that the whole manuscript should be accordingly changed for a more general objective, which should be the evaluation of the neuroprotective effects of dioscin.

4.       Why did the authors select PC12 cell line for the in vitro studies? This line derives from adrenal phaeochromocytoma, being mainly used for cathecolamine neurosecretory studies. Although it has some characteristics of neuron, only in presence of nerve growth factor it differenciates to sympathetic peripheral neuron. Please, discuss more extensively this point.

5.       Please, in material and methods, briefly indicate the rationale for evaluating every marker. It is very difficult to follow the manuscript, because the authors assume that the reader know exactly each role of every protein measured. In this context, no explanation of the objective for measuring MAP2 and GFAP.

6.       In point 3.5 (line 256), authors mention that the number of neurons were decreased in the model group compared to control rats. Nevertheless, there is no graph or figure showing such asseveration.

7.       Fig 1.F. is so tiny that makes nearly impossible to detect any morphological change in PC12 cells induced by H202.  Please, redo showing pictures with higher magnification.

8.       The resolution of Figure 4 is quite low and most important the hippocampal areas of each group is completely different. Thus, in some figures the dentate gyrus can be observed, but in others pictures, some CA area seems to be depicted. Please, in order to do a fair comparative, select pictures from the same hippocampal area.  In addition, quantifiy the signal from the microphotographs obtained.

9.       In all figures with cell or brain slices, insert a scale bar indicating the actual magnification. In line 172 I have seen a mistake regarding the magnification of the images obtained with transmission electron microscope. When it states 400x, it should be 40,000x (as pointed in Figure 5).

10.   Please, the results from electron microscopy should be more extensively explained. In  this sense, explain the basis of DCFH-DA model.

11.   Figure 6C. Quantity ROS levels and how dioscin was able to reduce it.

12.   For neuroinflammation, instead of GFAP (marker of astroglia) it should be more informative to carry out immunofluorescence of microglia marker such as Iba-1.

13.   Include in material and methods or in the appropriate section of the discussion the role of JNK, ERK, CYP2E1, IL-6, SOD, P38, etc, in neurodegeneration or aging processes.

14.   In section 3.2, authors state that dioscin has a potent hypoglycemic effect. Indeed, from the results dioscin has not hypoglycemic effect since it does lower the basal plasma glucose level. Instead, it can prevent the hyperglycemia induced by the model, therefore, dioscin show antihyperglycemic but not hypoglucemic effect.

15.   Authors establish the concentration of dioscin and H2O2 for the in vitro experiment. However, I did not find the rationale for choosing the different doses (20 to 60 mg/kg) for the in vivo experiment.

Author Response

Replaying to the comments from Reviewer 2:

The current manuscript entitled “Protective Effect of Dioscin on Brain Aging in Rats” by Qi and coworkers deals with the eventual neuroprotective effects of dioscin, a natural steroidal saponin found in several vegetal species, against brain aging. To this aim, the authors have used different experimental models both in PC12 cell culture and in rats. This paper is continuation of numerous previous works of the same research group, in which the authors have repeatedly shown a protective effect of dioscin in a wide range of pathologies. Although for the aforementioned reason, the novelty of this study is questionable, the topic of this study is an important one, since aging of population is one of the main challenges of the humanity for the next decades. Therefore, the quest for nootropic agents or brain aging delaying drugs is an intense subject in neuropharmacology nowadays

According to the results found by the authors, dioscin, both in vitro and in vivo, had powerful neuroprotective effects, preventing or reducing several neurochemical markers of brain damage, including ROS, expression of proteins involved in neuronal integrity, histological and ultrahistological markers and even having a noticeable impact on learning and memory. The number of experiments carried out by the authors in order to determine the beneficial effects of dioscin is outstanding and impressive. Finally, the authors conclude that dioscin-mediated effects are associated to a reduction of oxidative stress by normalizing the Nrf2/ARE signal pathway. I do really consider that the present study is well performed and soundness. Nevertheless, there are some concerns that the authors should address. My main concern is how the different models resemble the real senescence process.

Major comments:

1. Although the manuscript is understandable, the writing has too many mistakes and weird expressions. Therefore, I highly recommend to check the whole manuscript by an English native speaker.

Response: Thank you for the valuable comment. The grammar mistakes have been carefully corrected and the whole manuscript has been checked by an English native speaker.

2. Authors have to discuss the neuroprotective effect of dioscin under the light that several previous studies (some from your research group) have showed that dioscine promotes apoptosis and cell death. Those previous results pointed to dioscin as an eventual cytotoxic agent against multiple types of cancer.

Response: Thank you for giving us the valuable comment. Some researches of our group showed that dioscin has an eventual cytotoxicity against multiple types of cancer. But as shown in our research (Tao et al, 2017; Si et al, 2017), only high concentration of dioscin (μM grade) has cytotoxic effect, but the concentration used in this study is less than 400ng/mL (0.46 nM), which will not produce cytotoxic effect. Similarly, many nature products have the same effects such as resveratrol, which has cytotoxic effect at high concentration (Yousef et al, 2017) and neuroprotective effect at low concentration (Bastianetto et al, 2015).

Tao X, Xu L, Yin L, Han X, Qi Y, Xu Y, Song S, Zhao Y, Peng J. ioscin induces prostate cancer cell apoptosis through activation of estrogen receptor-β. Cell Death Dis. 2017, 8(8): e2989.

Si L, Xu L, Yin L, Qi Y, Han X, Xu Y, Zhao Y, Liu K, Peng J. Potent effects of dioscin against pancreatic cancer via miR-149-3P-mediated inhibition of the Akt1 signalling pathway. Br J Pharmacol. 2017, 174(7): 553-568.

Yousef M, Vlachogiannis IA, Tsiani E. Effects of Resveratrol against Lung Cancer: In Vitro and In Vivo Studies. Nutrients. 2017, 9(11). pii: E1231.

Bastianetto S, Ménard C, Quirion R. Neuroprotective action of resveratrol. Biochim Biophys Acta. 2015, 1852(6):1195-1201.

3. All studies carried out point to a general neuroprotective effect of dioscin, but not an anti-aging effect. Although, the D-galactose rat model is a well-recognized model of aging, resembling many of the features of aged rats, I really feel that the authors should have used normal aged rats to test the putative anti-aging effects of dioscin. Indeed, in order to claim antiaging actions of dioscin, a longitudinal study in normal animals  Moreover, the in vitro study (H2O2-induced damage in PC12 cell line) is not consider a model of aging at all. Rather, it is a model of cell damage mediated by oxidative stress, which is obviously is found in elderly but also in many other brain pathologies. For this reason, I think that the whole manuscript should be accordingly changed for a more general objective, which should be the evaluation of the neuroprotective effects of dioscin.

Response: Thank you for the valuable comment. As the opinion pointed out, although D-galactose-induced aging model is a well-recognized model of aging, but normal aged rats can better reflect the natural aging state and characteristics. We intend to use the natural aged rat model for the next study base on your comment.

In aging, brain is more vulnerable to oxidative stress. The purpose of this study was to investigate the neuroprotective effect of dioscin on brain aging induced by oxidative stress. For this purpose, oxidative damage models in vitro and in vivo were used in this study. In vitro, model of PC12 cells treated by H2O2 was used to mimic the oxidative damage of nerve cells and D-galactose-induced brain senescence model was used in vivo. It is known that when an exogenous dose of D-galactose is given beyond normal concentration, this can induce aging effects in brain by increasing oxidative stress, apoptosis and inflammation (Shwe et al, 2018; Sadigh- Eteghad et al, 2017). D-galactose-injected rodent models recapitulate many features of brain ageing and have been extensively applied to study the mechanisms of brain ageing. Based on your comment, we added discuss about this point in the Discuss section on page 13. Besides, we agreed to your comment that the object of this manuscript should be more general and we have corrected the title and content to focus on the neuroprotective effects of dioscin on aging brain.

Shwe T, Pratchayasakul W, Chattipakorn N, Chattipakorn SC. Role of D-galactose- induced brain aging and its potential used for therapeutic interventions. Exp Gerontol. 2018, 101:13-36.

Sadigh-Eteghad S, Majdi A, McCann SK, Mahmoudi J, Vafaee MS, Macleod MR. D-galactose-induced brain ageing model: A systematic review and meta-analysis on cognitive outcomes and oxidative stress indices. PLoS One. 2017, 12(8): e0184122.

4. Why did the authors select PC12 cell line for the in vitro studies? This line derives from adrenal phaeochromocytoma, being mainly used for cathecolamine neurosecretory studies. Although it has some characteristics of neuron, only in presence of nerve growth factor it differenciates to sympathetic peripheral neuron. Please, discuss more extensively this point.

Response: Thank you for the valuable comment. In this study, we have utilized rat pheochromocytoma cells (highly differentiated PC12 cells) as a model since they differentiate to resemble sympathetic neurons morphologically and functionally when cultured in the presence of nervegrowth factor (NGF) and share many common features with neurons (Shafer TJ et al, 1991). In addition, PC12 are easy to culture and conditional immortalized, which make it an ideal model to study neurotoxic mechanisms and neuroprotective drugs. In recent years, numerous neuroprotective drugs have been studied based on the oxidative stress injury model of PC12 cells (de Los Rios C et al, 2018). It was proved that PC12 cells can be used as a reliable model of neuropharmacological cells to study the neuroprotective effect of drugs in vitro.

Shafer TJ, Atchison WD. Transmitter, ion channel and receptor properties of pheochromocytoma (PC12) cells: a model for neurotoxicological studies. Neurotox -icology. 1991, 12(3):473-492.

de Los Rios C, Cano-Abad MF, Villarroya M, López MG. Chromaffin cells as a model to evaluate mechanisms of cell death and neuroprotective compounds. Pflugers Arch. 2018, 470(1):187-198.

5. Please, in material and methods, briefly indicate the rationale for evaluating every marker. It is very difficult to follow the manuscript, because the authors assume that the reader know exactly each role of every protein measured. In this context, no explanation of the objective for measuring MAP2 and GFAP.

Response: Thank you for the valuable comment. As what you pointed, it’s difficult to follow the manuscript with no explanation about the markers. The description of the rationale for evaluating every marker has been added in the manuscript. Please see the changes colored in red on page 5.

6. In point 3.5 (line 256), authors mention that the number of neurons were decreased in the model group compared to control rats. Nevertheless, there is no graph or figure showing such asseveration.

Response: Thank you for giving us the valuable comment. The original description was defective and has been corrected. Please see the changes colored in red on page 8.

7. Fig 1.F. is so tiny that makes nearly impossible to detect any morphological change in PC12 cells induced by H202. Please, redo showing pictures with higher magnification.

Response: Thank you for the valuable comment. Based on your comment, pictures with higher magnification (400 ×) have been selected to detect the morphological changes in PC12 cells induced by H202 as shown in Fig 1.F on page 7.

8. The resolution of Figure 4 is quite low and most important the hippocampal areas of each group is completely different. Thus, in some figures the dentate gyrus can be observed, but in others pictures, some CA area seems to be depicted. Please, in order to do a fair comparative, select pictures from the same hippocampal area. In addition, quantifiy the signal from the microphotographs obtained.

Response: Thank you for giving us the valuable comment. Figure 4 was redone to improve picture quality. In Figure 4, images of hippocampal CA area were selected to show the changes of MAP2 of neurons. Images of cortex area were selected to show the changes of GFAP of astrocytes and Iba1 of microglia. In addition, the images have been quantified. Please see Figure 4 on page 9.

9. In all figures with cell or brain slices, insert a scale bar indicating the actual magnification. In line 172 I have seen a mistake regarding the magnification of the images obtained with transmission electron microscope. When it states 400x, it should be 40,000x (as pointed in Figure 5).

Response: Thank you for giving us the valuable comment. Scale bars have been inserted in all microscope figures and the corresponding describes have been has been indicated in the figure legend (colored in red) based on your comment. In addition, the magnification of the images obtained with transmission electron microscope has been corrected to 40,000x on page 5 “2.14”.

10. Please, the results from electron microscopy should be more extensively explained. In this sense, explain the basis of DCFH-DA model.

Response: The description of TEM assay results has been explained more extensively. Please see the changes colored in red on page 9 and page 10. The explanation of DCFH-DA model and TEM was added on the discussion section of page 14.

11. Figure 6C. Quantity ROS levels and how dioscin was able to reduce it.

Response: Thank you for the valuable comment. The ROS levels have been quantized in Figure 6C and the description of how dioscin reduce it has been added. Please see the changes colored in red on page 11 and the discussion section on page 14.

12. For neuroinflammation, instead of GFAP (marker of astroglia) it should be more informative to carry out immunofluorescence of microglia marker such as Iba-1.

Response: Thank you for the valuable comment. The immunofluorescence of IBA1 level has been assayed and quantified based on your comment. And the results have been added into Figure 4. Please see the changes in manuscript colored in red on page 8 and page 14.

13. Include in material and methods or in the appropriate section of the discussion the role of JNK, ERK, CYP2E1, IL-6, SOD, P38, etc, in neurodegeneration or aging processes.

Response: Thank you for giving us the valuable comment. The role of JNK, ERK, P38, CYP2E1, HMGB1, IL-6 and SOD in brain aging was described in the discussion. Please see the changes in manuscript colored in red on page 14 and page 15.

14. In section 3.2, authors state that dioscin has a potent hypoglycemic effect. Indeed, from the results dioscin has not hypoglycemic effect since it does lower the basal plasma glucose level. Instead, it can prevent the hyperglycemia induced by the model, therefore, dioscin show antihyperglycemic but not hypoglycemic effect.

Response: Thank you for the valuable comment. The manuscript has been corrected based on your comment. Please see the changes colored in red on page 7.

15. Authors establish the concentration of dioscin and H2O2 for the in vitro experiment. However, I did not find the rationale for choosing the different doses (20 to 60 mg/kg) for the in vivo experiment.

Response: Thank you for the valuable comment. This study is the continuation and extension of our previous works (Tao et al, 2015; Yao et al, 2017). The appropriate doses have been screened in the previous studies, and the preliminary experiment also conducted in this study, so this doses range was selected for the in vivo experiment.

Tao X., Sun X., Yin L, Han X., Xu L., Qi Y., Xu Y., Li H., Lin Y., Liu, K., et al. Dioscin ameliorates cerebral ischemia/reperfusion injury through the downregulation of TLR4 signaling via HMGB-1 inhibition. Free Radic Biol Med. 2015, 84, 103–115.

Yao H., Sun Y., Song S., Qi Y., Tao X., Xu L., Yin.L., Han X., Xu Y., Li H., et al. Protective Effects of Dioscin against Lipopolysaccharide-Induced Acute Lung Injury through Inhibition of Oxidative Stress and Inflammation. Front Pharmacol. 2017, 8, 120.

Round 2

Reviewer 2 Report

The authors have satisfactorily responded to all the concerns I raised about the initial version of the manuscript “Neuroprotective Effect of Dioscin on Aging Brain" by Qi et al. Indeed, most of my queries have been appropiately resolved and clarified, even the authors have carried out Iba-1 immunofluorescence for evaluating activated microglia as a marker of neuroinflammation. Consequently, I have to congratulate the authors for this piece of research with such an impressive amount of data (behavioral, neurohistochemical and biochemical among others) bringing more light about dioscine as a putative neuroprotective and antiaging drug.

In my opinion, this revised version of the manuscript has greatly improved from the original version, therefore I consider that now it can be accepted for publication in Molecules journal.